# The Multitasker Protein: A Look at the Multiple Capabilities of NUMB

**DOI:** 10.3390/cells12020333

**Published:** 2023-01-15

**Authors:** Sara M. Ortega-Campos, José Manuel García-Heredia

**Affiliations:** 1Instituto de Biomedicina de Sevilla (IBIS), Hospital Universitario Virgen del Rocío (HUVR), Consejo Superior de Investigaciones Científicas, Universidad de Sevilla, 41013 Sevilla, Spain; 2CIBERONC, Instituto de Salud Carlos III, 28029 Madrid, Spain; 3Departamento de Bioquímica Vegetal y Biología Molecular, Universidad de Sevilla, 41012 Sevilla, Spain

**Keywords:** NUMB, isoforms, development, cancer, Alzheimer, neurogenesis

## Abstract

NUMB, a plasma membrane-associated protein originally described in *Drosophila*, is involved in determining cell function and fate during early stages of development. It is secreted asymmetrically in dividing cells, with one daughter cell inheriting NUMB and the other inheriting its antagonist, NOTCH. NUMB has been proposed as a polarizing agent and has multiple functions, including endocytosis and serving as an adaptor in various cellular pathways such as NOTCH, Hedgehog, and the P53-MDM2 axis. Due to its role in maintaining cellular homeostasis, it has been suggested that NUMB may be involved in various human pathologies such as cancer and Alzheimer’s disease. Further research on NUMB could aid in understanding disease mechanisms and advancing the field of personalized medicine and the development of new therapies.

## 1. Introduction

The study of the bases of development has been shown to be an effective way to understand the molecular basis of human pathologies. Investigating proteins that are involved in human diseases and play a role in development and cell fate could be key to making connections and advances by understanding how their presence and function modify cell behavior. One such multitasking proteins is NUMB [1], which has a wide range of functions in various species [2]. It is a cell membrane-associated protein that plays a crucial role in determining cell fate. Originally, NUMB was first described as an antagonist of the NOTCH membrane receptor in sensory organ precursor cells (SOPs) in *Drosophila melanogaster* [3]. It is secreted asymmetrically during cell division, producing two types of cells: one that retains the characteristics of the parent cell and one that is capable of differentiation [4,5].

NUMB is highly evolutionarily conserved. Since its discovery in *Drosophila*, two homologous proteins, NUMB and NUMBL, have been found in multiple mammals, although most of the research has been carried out in mice (mNumb/mNumbl) and humans (NUMB/NUMBL) [6,7]. NUMBL performs many functions that overlap with NUMB, although certain aspects of NUMBL remain to be understood [6]. NUMB is involved in important cellular processes, such as protein labeling for endocytosis, ubiquitination, cell adhesion, migration, and asymmetric cell division. Unlike NUMB, NUMBL is secreted symmetrically to daughter cells [7]. The role of NUMB as a cell fate determinant has been related to processes where there is a balance between self-renewal and cell fate determination [8,9,10,11,12,13,14,15]. This has led to its proposed involvement in pathologies related to the disruption of this balance, such as cancer or Alzheimer’s disease [16,17,18,19,20,21,22,23]. NUMB has been proposed to be involved in the regulation of multiple cellular pathways commonly altered in cancer, such as Wnt, Notch, or Hedgehog, due to its interactions with a wide variety of proteins [24,25,26,27,28]. It is also thought to be involved in the maintenance of the cancer stem cell (CSC) pool [28,29]. NUMB has also been linked to Alzheimer’s disease, where it plays a role as an adaptor of amyloid precursor protein (APP), an essential protein in the pathogenesis of the disease [23]. Recently, differential expression of NUMB has been proposed as a prognostic factor in various types of cancer [2,17,30,31,32,33]. It has been described that eukaryotic NUMB mRNA can undergo alternative splicing to produce at least four different isoforms at the protein level (called p72, p71, p66, and p65) produced by the inclusion or exclusion of exons 3 and 9, making NUMB an even more multifunctional protein [34,35,36]. This has led to the proposed prognostic value of each of them in some cases [34,37,38,39,40]. Because of its adaptor role, NUMB has also been proposed as a therapeutic target, not only in different types of cancer but also in various pathologies such as Alzheimer’s disease, among others [41,42,43,44,45,46].

In this work, we review the various roles of NUMB in the cell as an adaptor, polarization agent, and endocytic protein, and how these functions are involved in the development of pathologies such as Alzheimer’s disease and cancer. We also detail the possible mechanisms responsible for regulating NUMB and its potential as a target for the development of therapies for various diseases, as well as its usefulness as a biomarker.

## 2. *NUMB* Gene

*NUMB*, also known as S171, C14orf41, or C14-5527, is a gene located on the 14q24.3 chromosomal region that encodes an endocytic protein of the same name [47,48]. It is one of 19 different transcripts derived from genes in the AD3 region of chromosome 14, which is associated with familial Alzheimer’s disease [28,49]. This form of dementia develops slowly and progressively worsens, causing problems with memory, coordination, and thinking [50]. It is characterized by the aggregation of β-amyloid peptides, which are generated from APP by sequential cleavage, first by β-secretase and then by γ-secretase [51]. Although most cases of Alzheimer’s disease have a sporadic onset, familial Alzheimer’s disease is mainly caused by mutations in three genes: *APP*, presenilin 1 (*PSEN1*), and presenilin 2 (*PSEN2*) [52]. In fact, the AD3 region contains, among others, the *PSEN1* gene, which encodes the catalytic domain of γ-secretase that is also responsible for cleaving the NOTCH receptor [28,49].

*NUMB* is expressed in different adult human tissues, with the highest levels found in the blood, lung, and gallbladder and the lowest in the pancreas [48,53]. However, during development, *NUMB* expression varies in a stage-dependent manner, peaking at the two-cell embryo stage and gradually decreasing in later stages before blastocyst formation [54]. There are no known germline diseases due to single nucleotide polymorphism changes in the *NUMB* gene [55]. In addition, *NUMB* mutations are usually not point mutations, but rather amplifications or deep deletions [56]. However, little is known about the epigenetic status of *NUMB* in neurogenesis and development. The *NUMB* promoter appears to be hypermethylated in tumors, such as breast invasive carcinoma, lung adenocarcinoma, and colon adenocarcinoma, compared to normal tissue. On the other hand, its close homolog, *NUMBL*, appears to be hypomethylated in tumors such as colon and lung adenocarcinoma, compared to non-tumor tissue [6,57].

This gene is composed of nine exons that encode up to nine isoforms through alternative splicing. Only four of these isoforms encode a protein, which are named according to their molecular weight: p72 (variant 1), p66 (variant 2), p71 (variant 3), and p65 (variant 4) [35,58,59]. The longest transcript and isoform is p72, which includes all exons. Isoform p66 lacks exon 9, while isoform p71 lacks exon 3. The shortest variant is p65, which lacks both exons 3 and 9 (Figure 1) [35,48,59].

## 3. NUMB Protein

NUMB is a membrane-associated protein that is primarily found in the cell and peripheral membrane [48,53]. However, it is also believed to be present in the endosomal membrane, clathrin vesicles, the basolateral membrane, nuclei, and cytoplasm [58]. The N-terminal region of NUMB presents a phosphotyrosine binding (PTB) domain, while the C-terminal region presents an Asn-Pro-Phe (NPF) motif, which allows it to interact with the EH domains found in endocytic proteins [49,55,60,61]. In addition, NUMB has a domain located near the PTB domain, called NUMB-F, the function of which remains unknown [49,55]. NUMB also has a proline-rich region (PRR) that serves as an Src homology 3-binding domain. The PTB and PRR domains are affected by alternative splicing, as they include exons 3 and 9, respectively. As a result, the presence or absence of these exons can result in long or short PTB (PTB^L/S^) and long or short PRR (PRR^L/S^) domains in the isoforms [62,63] (Figure 1).

NUMB interacts with a variety of proteins, including REPS1 (RalBP1-associated Eps domain-containing protein 1), CTNNB1 (β-catenin 1), A4 (APP amyloid β precursor 4 protein), p53, and Mdm2/HDM2 (Murine/Human Double Minute 2) [39,48,58]. Furthermore, the interaction of NUMB with the oncogenic protein MAP17 leads to the aberrant activation of the NOTCH pathway and an increase in tumorigenic cell properties [28]. NUMB can also be post-translationally modified, mainly through phosphorylation and methylation, which can alter its ability to interact with other proteins. For example, NUMB phosphorylation at the Ser276 and/or Ser295 residues by CAMK-1 protein reduces the stability of one of the NUMB interactors, p53, by disrupting the NUMB-p53 interaction [64,65,66]. NUMB methylation by SET8 at Lys158 and Lys163 also causes dissociation of its interaction with p53 [67]. There are also isoform-specific modifications, as only p72 and p66, which contain the full PTB domain, can be ubiquitinated by the Ligand of the NUMB X (LNX) protein for subsequent degradation in the proteasome [53]. These facts make the NUMB interactome a very complex and dynamic network.

## 4. NUMB Is Involved in Asymmetric-Division Related Pathologies

Asymmetric division plays a vital role in generating cell diversity [68]. This phenomenon was first found in *Drosophila* and *Caenorhabditis elegans*, where cell fate proteins are passed asymmetrically to daughter cells in precursor cell division [69]. The SOP in *Drosophila* is crucial for the development of the central and peripheral nervous system [68,70]. It undergoes rounds of asymmetric division to produce four daughter cells: sheath, neuron, socket, and hair [68,70]. NUMB is secreted in one of the daughter cells (anterior pI daughter cell b, pIIb), while its antagonist, NOTCH, is secreted in the other daughter cell (pIIa) [68,70,71]. This is due to NUMB being polarized to one side of the cell during mitosis, demonstrating its role as a cell fate determinant (Figure 2A) [68,70,71]. NUMB and NOTCH also control asymmetric division in neuroblasts, which inherit NOTCH, and neural stem cells (NSCs), also called ganglion mother cells, which inherit NUMB (Figure 2B) [72,73]. Par6, Baz, and aPKC, apical polarity proteins, are located on one side of the neuroblast, while Lgl (Lethal giant larvae), Dlg (Discs large), and NUMB are located at the basal pole [72,74]. The partition defective complex (Par3-Par6-aPKC) leads to polarized organization and asymmetric segregation of NUMB [75]. Additionally, Lgl is required for the proper asymmetric segregation of NUMB into daughter cells, as it promotes the formation of a basal crescent form of this protein [74]. However, proper positioning of NUMB and the orientation of the mitotic spindle also depend on the presence of the Inscuteable protein (Insc), a key component of the asymmetric segregation machinery in *Drosophila* [76,77]. Insc is located in the apical cortex of the cell before and during neuroblast mitosis, and it must interact with Bazooka to maintain the apical-basal polarity necessary for NUMB and other proteins to be asymmetrically segregated in metaphase [78]. Therefore, the relationship between apical-basal polarity and asymmetric segregation has been demonstrated [78]. Because NUMB interacts with other proteins through its PTB domain, it has been classified as a polarity marker [79,80,81]. 

The role of NUMB in maintaining neural progenitors has been well-stablished [72]. Both NUMB and NUMBL have been suggested to be involved in maintaining highly polarized radial glial cells and in cortical neurogenesis [72,82,83]. As a result, deleting NUMB and NUMBL in mice is embryonically lethal [84]. Therefore, to study the effects of NUMB and NUMBL deficiency in mice, a conditional NUMB–NUMBL knock-out model was created using Emx1-Cre (expression was induced on Day 9.5, avoiding early embryonic lethality). This resulted in disruption of the neuroepithelium, severe hydrocephalus, delayed cell cycle exit, impaired neural differentiation, and progenitor hyperproliferation [72,82].

Furthermore, NUMB has been shown to play an important role in the functions of the Par complex (as mentioned above) and cell-cell junctions, both processes commonly associated with epithelial-mesenchymal transition (EMT) [85,86,87]. In this process, epithelial cells lose their cell polarity and cell-cell adhesion, gaining both migratory and invasive properties to become mesenchymal stem cells [85,86,87].

NUMB has also been described as an essential agent for maintaining cell packing density during the elongation process of the mammary duct epithelial tube. NUMB loss caused an aberrant distribution of E-cadherin, leading to cells with lower tension, altered shape, and increased packing, resulting in a reduction in duct elongation [88]. Additionally, NUMB and NUMBL, in cooperation with sarcomeric α-actin, have been found to be essential for regulating Z-disc consolidation in sarcomere assembly and its maintenance in striated muscle [89,90]. 

On the other hand, EMT is also involved in several pathologies, such as endometriosis, where NUMB down-regulation is correlated with increased cell migration and invasion [91]. A more representative example is its role in tumorigenesis, where NUMB and NOTCH establish a delicate balance that, if disturbed, can lead to aberrant differentiation and cancer progression and metastasis (Figure 2C) [92,93]. Specifically, an increase in the number of asymmetric divisions has been identified as the starting point for the development of cancer stem cells. In more detail, it has been proposed that NUMB isoforms with PRR^L^ may be involved in the early stages of cancer development, promoting proliferation, while NUMB isoforms with PRR^S^ may be involved in the latter stages of cancer, inducing differentiation and loss of cell polarity [94]. Furthermore, NUMB overexpression appears to regulate the malignant transition through the regulation of different pathways. It has been shown to promote EMT through TGFB-dependent ZEB1/Snail2 and MAPK signaling in pancreatic cancer, but it has also been found to inhibit EMT in tongue cancer through RBP-JK-dependent NOTCH1/PTEN/FAK signaling, PAK1/β-catenin signaling in ovarian cancer, and through WNT in colorectal cancer [95,96,97,98,99,100]. 

NUMB may also play a role in regulating cell adhesion and polarity in response to tyrosine kinase signaling [101]. Interestingly, there is evidence to suggest that in many types of tumors, the EMT transition may be caused by abnormal activity in this pathway [85,101]. When NUMB was knocked down in MDCK cells, it led to the delocalization of the Par3 complex and aPKC, as well as the apical-basal translocation of E-cadherin and β-catenin, polymerization of F-actin, and a decrease in cell-cell adhesion, resulting in an increase in cell proliferation and migration [85,101]. 

## 5. NUMB Maintains Cellular Homeostasis by the Regulation of the Endocytic Machinery

NUMB has been suggested to play a role in maintaining cellular equilibrium, specifically in the endocytosis process [102]. This was suggested by the discovery of the interaction of the NPF NUMB motif with Eps15, a component of the endocytic machinery, through its EH domain [2,102,103]. Eps15 is involved in the transport and sorting of molecules [60,104]. However, the deletion of the NPF NUMB motif does not appear to affect NUMB functions, leading to the possibility that its endocytic role may follow an independent proteasome pathway that could play a major role in determining cell fate [105,106,107]. Both Eps15 and the AP-2 adaptor complex are involved in clathrin-mediated endocytosis [105]. Interestingly, Eps15 and the three subunits of the AP-2 adaptor complex appear to interact more strongly with NUMB isoforms containing exon 9 [106]. These interactions allow NUMB to function as a protein involved in the endocytic machinery [2]. 

Clathrin-dependent endocytosis is a process in which extracellular fluid and proteins are mixed and packaged into clathrin-coated vesicles [68]. NUMB appears to be involved in the localization and co-trafficking of endocytic organelles, as well in the endocytosis of internalized receptors [102]. In *Drosophila*, NUMB segregation in one of the SOP daughters that results in increased endocytosis and inhibition of NOTCH signaling [61,108,109]. As mentioned above, NUMB is also required for the establishment of NOTCH signaling during cytokinesis [110]. In dividing cells, NUMB delocalizes from the basal cortex of pIIb in a process dependent on the protein α-adaptin [110,111,112]. The interaction between NUMB and the Ear domain of α-adaptin, a subunit of the AP-2 complex, leads to the preferential secretion of α-adaptin in pIIb cells during asymmetric division [111,112]. NUMB appears to act as an adaptor, allowing AP-2 to bind to the NOTCH intracellular domain (NICD) on one side of the pIIb cell. This results in the internalization of the NOTCH receptor and decreased NOTCH activity in pIIb cells [112,113,114]. Through this mechanism, NUMB acts as an inhibitor of the NOTCH pathway through the polarized endocytosis of the NOTCH receptor, while also serving as a cell fate regulator through its binding to NICD [112,115]. Endocytosis is also thought to be critical for the balance between self-renewal and differentiation in NSCs [116]. In this process, NUMB interacts with α-adaptin through the Trunk domain of α-adaptin, regulating the behavior of NSCs through NOTCH. However, this interaction occurs through a different domain of α-adaptin than the one involved in NUMB/α-adaptin interaction in SOPs [116].

In addition, the four-pass transmembrane protein Sanpodo (Spdo) is also involved in NOTCH signaling, which determines NUMB-mediated cell fate [108]. SPDO is internalized during cytokinesis [108] and accumulates with NUMB in pIIb cells, interacting with the NUMB PTB domain through its NPAF (Asn-Pro-Ala-Phe) motif [110,117,118]. In plla cells, SPDO is found on the cortical surface, but in pllb cells, it colocalizes with NOTCH and Delta in RAB5/RAB7-positive endocytic vesicles [119]. NUMB has been proposed to interact with NOTCH-SPDO oligomers in early endosomes, inhibiting NOTCH recycling and allowing for asymmetric distribution of NOTCH on the surface of pIIa/b daughter cells, thereby regulating cell fate determination [110,117]. NUMB also promotes SPDO targeting from endosomes to the plasma membrane, possibly due to NOTCH inhibition [111,118,120]. NUMB is responsible for SPDO removal from the membrane, and its internalization is incompatible with productive NOTCH signaling in pIIb cells. However, in pIIa cells, where NUMB is absent, SPDO remains in the membrane [121]. 

NUMB has been proposed to act as a regulator of the balance between NOTCH recycling and targeting to late endosomes in neural progenitor cells in *Drosophila* [122]. Interestingly, NUMB regulates NOTCH trafficking to RAB7-labeled late endosomes, but not to early endosomes [122]. In mammals, NOTCH1 is also constitutively internalized, with differences in its trafficking dynamics depending on changes in NUMB expression [113]. NUMB can also inhibit NOTCH1 activity by regulating post-endocytic sorting events that lead to the degradation of NOTCH1, redirecting the protein to the late endosome compartment [113]. However, NUMB/NUMBL also appears to play a role in sensory axon arborization in neurons by regulating NOTCH1 through the endocytic-lysosomal pathway. In mice, conditional deletion of *NUMB* in a *NUMBL* null background resulted in reduced endocytosis and a decrease in axon branch points [83]. It also resulted in a reduction of overall axon length, likely due to the accumulation of NOTCH1 in nuclei [123]. Mammalian NUMB can also antagonize the NOTCH pathway by controlling the post-endocytic trafficking of the NOTCH ligand Delta-like 4 (DLL4) [124]. Low NUMB levels lead to the accumulation of DLL4 on the cell surface, causing aberrant activation of the NOTCH pathway. The NUMB/NUMBL knockdown model showed impairment in the targeting of DLL4 to lysosomes, allowing it to be recycled from the cell surface by RAB11-positive endosomes [124].

NUMB functions as an endocytic regulator of various adhesion molecules, such as integrins and E-cadherin [88]. Clathrin-dependent endocytosis of integrins is a widely recognized process that is essential for cell migration [125]. NUMB binds to β-integrins and colocalizes with them to clathrin-coated structures (CCSs). NUMB phosphorylation by aPKC results in its release from CCSs, preventing it from binding to integrins [126,127]. This inhibition of binding may contribute to cell migration through NUMB’s interaction with Par-3, directing integrin endocytosis to the leading edge [126]. On the other hand, NUMB can also control cadherin-based adhesion through its interaction with p120 catenin [128], a protein that inhibits E-cadherin internalization. Phosphorylation of NUMB by aPKC can prevent its association with p120, attenuating E-cadherin endocytosis and maintaining apicobasal polarity [128]. In addition, up to 25 Ser/Thr phosphorylation sites have been described [129], suggesting a complex regulation of NUMB functions.

As mentioned above, NUMB acts as a cargo-selective endocytic adaptor protein by binding other proteins to the clathrin α-adaptin adaptor [130,131]. This binding is regulated by NUMB phosphorylation, specifically at Ser265 and Ser284, which promotes the recruitment of the 14-3-3 protein. This causes NUMB to dissociate from α-adaptin and translocate from the cortical membrane to the cytosol [130]. NUMB can also be phosphorylated at Ser283, after initial phosphorylation at Ser264, by Ca^2+^/calmodulin-dependent protein kinase, which abolishes the binding of AP-2 to NUMB and promotes the NUMB-14-3-3 interaction [132,133]. Ca^2+^/calmodulin-dependent protein kinase may also disrupt the NUMB/AP-2 interaction by phosphorylating NUMB at Ser276 [134]. NUMB phosphorylation by AAK1 (adaptor-associated kinase) appears to be essential for reducing the clathrin coat [135]. Furthermore, CDK5 can phosphorylate NUMB at Ser288, which may play a role in the RAC/RHO axis for controlling cell adhesion and migration [129]. 

Interestingly, NUMB isoforms appear to regulate the internalization of mGluR5 and mGluR1 differently [136,137,138]. These two metabotropic glutamate receptors are responsible for synaptic development and emotional and motor behaviors in the central nervous system [136,137,138]. The p72 isoform, but not p65, binds to mGluR5 or mGluR1, and increases its expression in the neuronal membrane by inhibiting their endocytosis [136,137]. Additionally, NUMB also regulates the different responses of neural progenitors in an isoform-dependent manner through the regulation of Ca^2+^ channels [139]. The NUMB PTB^L^ isoforms (p72 and p66) are closely associated with the endocytosis of Ca^2+^ channels, resulting in their accumulation inside the cell. In contrast, the PTB^S^ isoforms (p71 and p65) preferentially localize the channels to the membrane [139]. NUMB isoforms also appear to have differential role in the NSCs of the brain of *Drosophila* larvae [36]. PRR^L^ isoforms are expressed during early neurogenesis and promote proliferation, while PRR^S^ isoforms are expressed during neurogenesis and inhibit stem cell proliferation while promoting differentiation. This differential function in different phases of neurogenesis results from endocytic degradation [36].

The presence of enlarged endosomes is an early feature of Alzheimer’s disease and indicates deregulated endocytosis [140]. Furthermore, 70% of the β-amyloid peptide secreted in interstitial fluid is generated through processes related to endocytosis [140,141]. It has been proposed that NUMB may play a role in APP trafficking in an isoform-dependent manner. The PTB^L^ isoforms of NUMB appear to be responsible for targeting APP to late endosome/lysosome, while PTB^S^ isoforms may be responsible for APP accumulation in early endosomes [23]. The C-terminus of the APP family, specifically the YENTPY (Tyr-Glu-Asn-Thr-Pro-Tyr) domain, acts as an anchor for many proteins involved in clathrin-mediated endocytosis and exhibits increased affinity for NUMB PTB^L^ isoforms [23,140,142]. The collapsin response mediator protein 2 (CRMP2) is involved in NUMB-mediated endocytosis, and its increased phosphorylated state is considered an early sign of Alzheimer’s disease [133,143]. CRMP2 colocalizes with NUMB in the central region of axonal growing cones in the neurons of the hippocampus. The NUMB–CRMP2 interaction is mediated by the NUMB-PTB domain, which regulates NUMB-dependent endocytosis in the growth cone [133]. CRMP2 is also involved in NUMB-dependent endocytosis of other proteins, such as the neural L1 cell adhesion molecule (L1CAM), which is endocytosed and recycled in the growth cone, where NUMB and CRMP2 are located [144]. 

In cancer, NUMB is involved in the non-random segregation of subcellular vesicles [96]. Furthermore, NUMB plays a role in regulating the endocytosis of ALK (anaplastic lymphoma kinase), a receptor that is often aberrantly expressed in cancer [145,146,147]. Interestingly, this regulation appears to be isoform-specific; both NUMB PTB^L^ isoforms can promote receptor endocytosis, but p66 promotes ALK lysosomal degradation through RAB7-containing late endosomes, while p72 allows the kinase to remain active by promoting its recycling back to the plasma membrane [145].

## 6. NUMB Acts as an Adaptor Protein in the Multiple Signaling Pathways Involved in Morphogenesis Processes and Cancer Development

NUMB has been classified as a cargo-selective adaptor and is involved in several important cellular pathways (Figure 3) [24,25,26,27,28,70]. NUMB promotes the ubiquitination of NOTCH on the membrane by recruiting ITCH, which is a ubiquitin E3 ligase that promotes NICD degradation [26,148,149]. This prevents NICD from translocating into the nucleus and the subsequent transcription of NOTCH target genes [26,148,149]. The four NUMB isoforms appear to negatively regulate the transcriptional activity of NOTCH1, but not NOTCH2 or NOTCH3 [148]. Furthermore, NUMB and NOTCH are also inversely expressed during the progression of oligodendrocyte differentiation, with higher NUMB expression in mature oligodendrocytes [150]. In the developing neocortex, NUMB and NOTCH are expressed in the ventricular zone of progenitors, while NUMBL is expressed in postmitotic neurons in the cortical zone [7]. NUMB also plays a role in the proliferation of cardiomyocytes and trabecular morphogenesis through its interaction with NOTCH1 [151]. However, NUMB/NUMBL can inhibit NOTCH2 signaling to control heart myocardial compaction [152]. On the other hand, PRR^S^ isoforms have been suggested to suppress NOTCH signaling in lung cancer cells, while PRR^L^ isoforms increase it [153,154]. There is also a significant inverse correlation between NOTCH1 and NUMB expression in non-small cell lung cancer (NSCLC) [153,154].

In progenitor cells, NUMB has been described as a target of canonical WNT signaling. When activated, the canonical WNT signaling pathway activates the β-catenin cascade, which leads to the induction of NUMB expression. NUMB acts to inhibit NOTCH in progenitors, thereby promoting cell differentiation [24]. The NUMB-mediated WNT-NOTCH network is regulated by androgen receptors [155]. Its disruption has been found to play a role in several types of cancer, such as breast and colorectal cancer [24,97]. NUMB also serves as a key inhibitor of the Hedgehog pathway [156]. Thus, in early brain granule progenitor cells, NUMB overexpression leads to the inhibition of GLI1, which results in the inhibition of stem cell growth and self-renewal, and the promotion of cell differentiation [25,157]. Increased expression of SMO, a key component of the Hedgehog pathway, has been found to increase both the number of stem cells and the spread of chronic myeloid leukemia by decreasing NUMB levels in patients. However, in the SMO KD cell model, NUMB levels are increased, leading to stem cell depletion and slowing the spread of the disease [158]. Decreased NUMB levels have also been found to increase the castration-resistant population in prostate cancer cells as a result of the deregulation of the NOTCH and Hedgehog pathways [159]. It is important to note that NUMB and NUMBL exhibit different behaviors in the regulation of the pathways involved in stem maintenance. While NUMB is an activator of the WNT and Hedgehog pathways, NUMBL acts by inhibiting both pathways. However, both proteins inhibit NOTCH signaling [6]. 

In parallel, NUMB regulates p53 function by forming a tricomplex with both p53 and the E3 ubiquitin ligase MDM2. This prevents the ubiquitination and subsequent degradation of p53, resulting in elevated levels of p53 [19,160,161,162]. For MDM2 inhibition, the presence of exon 3, present in PTB^L^ isoforms, appears to be required [39,104,163]. Dysregulation of the NUMB/p53/MDM2 complex has been implicated in several tumorigenic events, including the initiation of kidney, breast, or pancreatic cancer [64,164,165]. However, the binding between NUMB and p53 is not only important in cancer, as NUMB also enhances asymmetric mammary stem cell divisions through its interaction with p53 [166].

## 7. Regulation of NUMB Expression

NUMB expression can be regulated by miRNAs [167,168,169]. In *Drosophila*, Bantam miRNA controls cell proliferation by inhibiting *NUMB* in order to bypass cell growth control and regulate the feedback process to maintain the robustness of the NOTCH pathway, which is essential for the fate and self-renewal of NSCs [170,171]. In humans, *NUMB* mRNA is often targeted by miR-146 family genes, which can affect tissue differentiation and contribute to the development of various diseases [168]. MiR-146a appears to regulate *NUMB* by influencing the balance between symmetric and asymmetric cell division. In colorectal cancer, it directs symmetric division by suppressing *NUMB* [172]. This miRNA also plays a role in oral carcinogenesis, promoting cancer cell proliferation and migration by targeting *NUMB*, *IRAK* and *TRAF6* [173]. However, miR-146a has been shown to have effects beyond tumorigenesis, including skewing the balance between muscle differentiation and cell proliferation by negatively regulating *NUMB* [174]. Furthermore, miR-146a down-regulation and subsequent NUMB overexpression have also been linked to the suppression of apoptosis and promotion of autophagy in chondrocytes in osteoarthritis and inflammation in the active phase of thyroid-associated ophthalmopathy [175]. MiR-146b has also been shown to promote carcinogenesis in neuroblastoma by targeting *NUMB* [176]. 

The miR-31/96/182 families are frequently up-regulated in tumors, such as head and neck squamous cell carcinoma (HNSCC), colorectal or prostate cancer. Therefore, upregulation of these miRNAs in HNSCC increases cell invasiveness and migration by targeting *NUMB* [167]. Specifically, miR-31, which targets *NUMB*, has been shown to promote carcinogenesis in colorectal cancer [177]. Furthermore, miR-9-5P, has also been shown to increase stem cell growth and metastasis in prostate cancer by negatively regulating *NUMB* [178]. Alternative splicing of NUMB isoforms can be indirectly modulated by miR-335, which targets the splicing factor *RBM10*. In tumors, overexpression of miR-335 has been linked to increased tumor growth and decreased expression of *RBM10*, as well as increased expression of long NUMB isoforms (p72/71) [179]. 

NUMB also appears to be involved in other pathologies, such as preeclampsia, a pregnancy disorder associated with an increased risk of neonatal, fetal, or maternal morbidity/mortality. Increased levels of miRNA-524-5p in this pathology are involved in the regulation of trophoblast proliferation and invasiveness by targeting *NUMB* and subsequent regulation of the NOTCH pathway [180].

## 8. NUMB as a Therapeutic Tool for Various Pathologies 

Recent studies have suggested NUMB as a possible biomarker for prognosis and/or response to certain therapies for cancer, Alzheimer’s disease, and other pathologies, with potential clinical applications [21,140,181,182,183,184]. 

Interestingly, there is controversy regarding the role of NUMB as a tumor suppressor, as oncogenic behavior has been observed in some cases [185]. NUMB downregulation has been linked to poor prognosis in various types of carcinomas, melanoma, and glioblastoma, among others [95,97,98,159,164,183,185,186,187,188,189,190,191,192]. On the other hand, NUMB up-regulation has been associated with poor prognosis in certain tumors, such as hepatocellular carcinoma (HCC) [193] and esophageal squamous cell carcinoma (ESCC), where NUMB overexpression has been linked to increased tumor recurrence and poor overall survival [194], but more specifically, the possible functional diversity of the isoforms has been investigated. In these carcinomas, p72/71 is often found to be downregulated. In both HCC and ESCC, the expression of p72/p71 NUMB isoforms has been associated with increased early recurrence and lower overall survival after surgery due to increased proliferation, migration, and invasion in cancer cells. On the other hand, higher p66/p65 expression levels promote the opposite effects [193,194,195,196]. Therefore, in these carcinomas, a difference in prognostic meaning can be made between the isoforms [193,194,195,196]. 

However, the role of NUMB appears to be tissue-dependent and the prognostic significance of the isoforms cannot be extrapolated to all tumors. For example, in medulloblastoma the opposite effect occurs, with p72/71 being upregulated and p66/65 downregulated [38]. 

In addition, NUMB has become a predictive biomarker for patients for whom therapy would be appropriate, for example, in patients with prostate cancer who could benefit from NOTCH inhibition or therapies that restore p53 function, such as the Nutlin-related class of anti-MDM2 inhibitors, which are already used in breast cancer [41,189].

NUMB overexpression has been associated with increased sensitivity to cisplatin treatment in patients with epithelioid malignant pleural mesothelioma, but with poor response to treatment in ESCC, suggesting a role for NUMB in resistance to therapy [194,197]. Thus, NUMB downregulation has been associated with increased castration-resistant progenitors in prostate cancer, resistance to imatinib in chronic myeloid leukemia, and increased radioresistance in pancreatic cancer. This last effect was reversed by NUMB upregulation through metformin treatment [159,198]. Inhibition of NOTCH/NUMB signaling has also been associated with increased radiation sensitivity in nasopharyngeal carcinoma [199]. In contrast, NUMB overexpression in NSCLC has been associated with increased sensitivity to radiation [200]. 

Regarding Alzheimer’s disease, pharmacological modulation of APP by downregulating NUMB has been proposed as a novel therapeutic strategy. The effect reduced the cleavage of APP by γ-secretase, subsequently reducing β-amyloid peptide levels [201]. Furthermore, the proposal of NUMB as a possible therapeutic target for Alzheimer’s disease has been taken even further, with the possibility that the therapeutic value of the isoforms could be differentiated. The switch from p72/66 to p71/65 isoforms is an essential step in increasing the accumulation of β-amyloid peptide plaques in this disease [202]. Therefore, NUMB could represent a potential therapeutic target for decreasing the accumulation of these peptides [202]. In addition, NUMB appears to be involved in an isoform-specific manner in the regulation of Tau protein levels, which is also implicated in this pathology. Only the overexpression of p72 is able to decrease intracellular Tau levels, enhancing neuronal electrical activity. Consequently, this isoform could be considered as an important therapeutic factor [46].

Finally, NUMB has also been proposed as a possible biomarker or therapeutic target for other pathologies. In the renal field, it seems to be related to diabetic nephropathy, kidney fibrosis, acute kidney injury, and proteinuric diseases [43,182,203,204,205]. In the case of renal fibrosis, NUMB appears to be overexpressed in affected kidneys compared to healthy kidneys, making it a potential biomarker for the disease [182,206]. Conversely, in acute kidney disease, NUMB promotes the activation of p53-mediated protective autophagy, making it a potential therapeutic target for this disease [43,205,207]. Similarly, NUMB has emerged as a promising therapeutic target for proteinuric pathologies due to its protective role against endoplasmic reticulum stress-associated apoptosis in these diseases [203].

Furthermore, NUMB inhibition has also been proposed as a possible effective therapeutic strategy for another type of fibrosis, lung fibrosis. In this disease, NUMB prevents the activation of β-catenin signaling through its interaction with casein kinase 2 [181]. 

In terms of cardiovascular pathologies, NUMB has been proposed as an indicator of an increased risk of coronary artery disease. In this case, lower NUMB expression indicates a higher risk of developing the disease [208]. In addition, NUMB has also been proposed as a therapeutic candidate for muscle and cardiac regeneration, as well as for congenital heart disease [209,210]. Therefore, NUMB downregulation has been classified as a potential strategy to inhibit ischemia-induced apoptosis [211].

## 9. Conclusions

The multitasking role of NUMB demonstrates its importance in maintaining cellular homeostasis. NUMB has been characterized as an essential protein for cell polarization and asymmetric division as well as for other processes, such as endocytosis and the regulation of multiple cellular pathways. Dysregulation of its function could be a cause of pathologies such as Alzheimer’s disease or cancer, further emphasizing its importance in the cell. However, much is still unknown about NUMB and its isoforms, which increases its potential for a variety of functions. Further study of NUMB and its isoforms could increase our knowledge of cellular mechanisms and the molecular causes of different pathologies, potentially leading to advances in personalized medicine.

## Figures and Tables

**Figure 1 cells-12-00333-f001:**
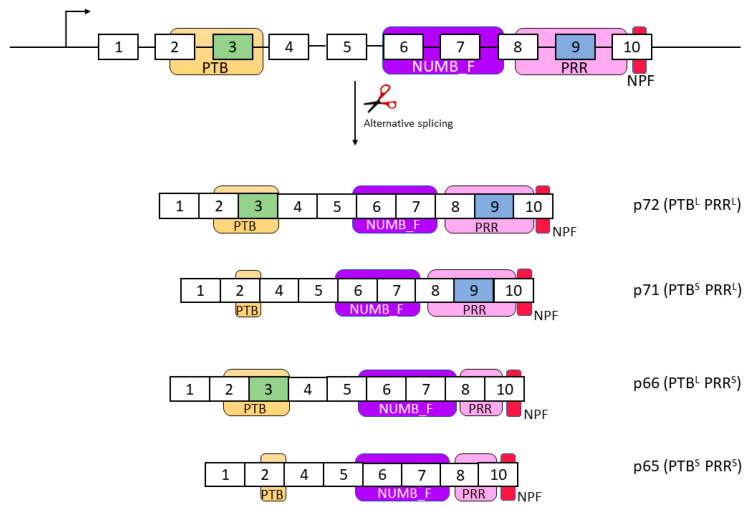
NUMB isoforms are generated by alternative splicing, being the PTB and PRR domains involved in this process. The resulting isoforms can be classified as having long or short PTB (PTB^L/S^) and long or short PRR (PRR^L/S^) domains. Please note that the sizes of exons and domains in the diagram are not to scale and are simplified for clarity.

**Figure 2 cells-12-00333-f002:**
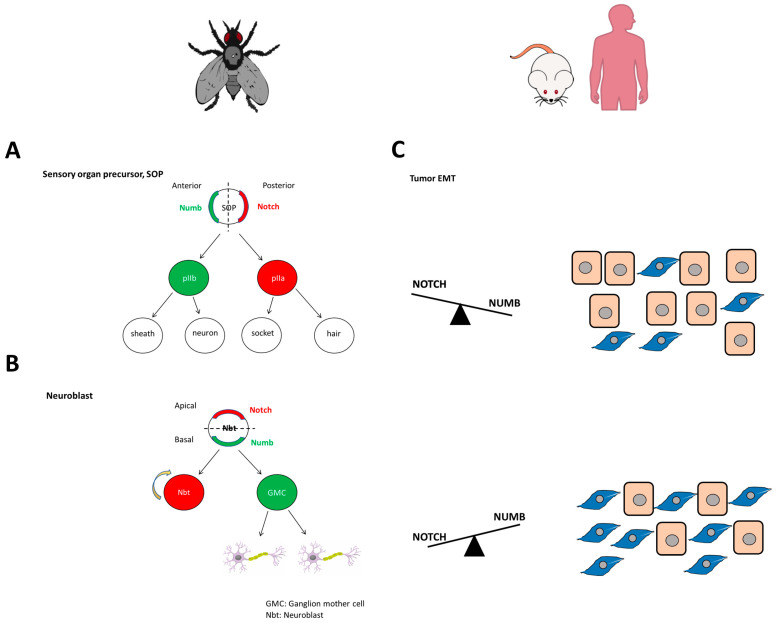
(**A**) NUMB and NOTCH are asymmetrically distributed to daughter cells during SOPs in *Drosophila*. (**B**) NUMB and NOTCH are involved in determining cell fate in *Drosophila* neuroblasts. The cell that inherits NOTCH (Neuroblast) has self-renewal properties, while the cell that inherits NUMB (GMC: ganglion mother cell) is able to differentiate into neurons. (**C**) The balance of NUMB and NOTCH is also thought to be involved in EMT in mice and humans tumors. Increased NUMB expression relative to NOTCH has been shown to reduce EMT, while increased NOTCH levels can increase EMT.

**Figure 3 cells-12-00333-f003:**
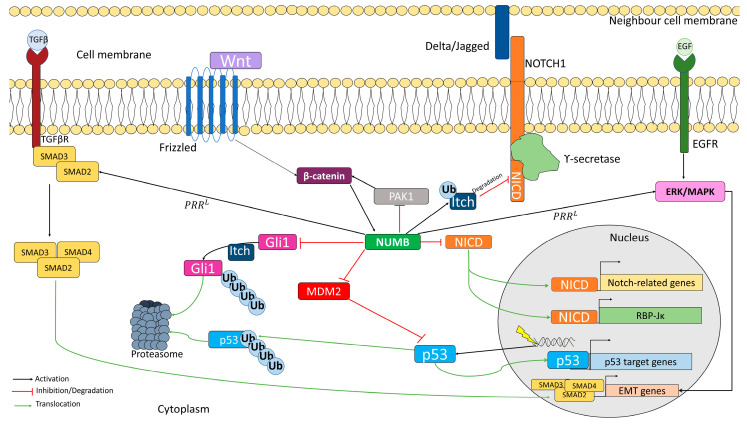
NUMB acts as an adaptor protein in multiple signaling pathways. NUMB inhibits NOTCH signaling by interacting with NICD. Additionally, the interaction between NUMB and MDM2 increases p53 stability, allowing transcription of genes related to DNA-damage repair. Furthermore, the PRR^L^ NUMB isoform promotes ERK/MAPK signaling and the formation of a SMAD complex, leading to transcription of genes related to EMT. In addition, NUMB promotes the ubiquitination and subsequent degradation of both NUMB and ITCH through GLI1.

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
