# Peer review of "The Multitasker Protein: A Look at the Multiple Capabilities of NUMB"

_cells, 2023, doi:10.3390/cells12020333_

Round 1

Reviewer 1 Report

Comments to the Author(s)

Title:The multitasker protein: A look at the multiple capabilities of NUMB.

Advantages: The role of NUMB proteins in participating in disease occurrence has been extensively studied, and numerous papers have reported the versatility of this protein. This manuscript will certainly increase our further understanding of NUMB protein function and enhance our knowledge of the possible functions and mechanisms of this protein in disease treatment. As an important research topic, the authors apparently spent a lot of time finding relevant literature. Meanwhile, authors summarize the contents of the paper in a graphic form, which is more intuitive to readers. However, the titles of the paper mainly focus on NUMB. If the NUMB content and disease content can be reasonably allocated and the graphics can be further modified, the paper will be better.

I recommend that the manuscript can be published after extensive revision.

Comments needed to be focused on:

1. This article provides a large introduction to NUMB, interspersed with an introduction to the functions of NUMB in cancer and Alzheimer's disease. However, a systematic review of the role and therapeutic potential of NUMB in cancer and Alzheimer's disease is not presented.

2. Because of the broad range of cancer inclusion, the authors feel intuitively fragmented, noncompact, and unsystematic in their introduction to the relationship of NUMB to various carcinomas. With the amount of literature allowed, the scope of cancer can be further narrowed at the time of writing.

3. Almost all titles are centered on NUMB, but the content under the title also contains a large number of descriptions of the relationship between NUMB and diseases. Therefore, some titles may be further modified.

4. Line 13  According to the previous literature, I think the author cannot support this statement.

5. Line 158  The word "Thus" is inappropriate.

6. The meanings of arrows in Figure 3 shall be marked, and the key steps shall be indicated with symbols.

7. Lines327-329  It was written in the original that NUMB promoted NOTCH ubiquitination and NICD degradation by recruiting ITCH, but this relationship was not directly reflected in Figure 3.

8. Lines 336-337  The role is not specific.

9. Line 337  The word "Otherwise" is inappropriate.

Author Response

Dear Reviewer,

First of all, we thank you for your time and effort in reviewing our work. We greatly appreciate the positive points you highlighted on how we presented our topic to the readers. Your suggestions for improvement have been very helpful. The authors have considered each of your comments and tried to satisfy each one of them until we achieved the manuscript that we now present. We hope that after this in-depth revision we will meet your standards. We would be pleased to receive further comments that will allow us to improve this work.

Below we provide responses to each of your comments.

  1. This article provides a large introduction to NUMB, interspersed with an introduction to the functions of NUMB in cancer and Alzheimer's disease. However, a systematic review of the role and therapeutic potential of NUMB in cancer and Alzheimer's disease is not presented.

Following your advice, we added a paragraph presenting the possible role of NUMB as a therapeutic tool in cancer and Alzheimer's disease to the Introduction (lines 49-57).

  1. Because of the broad range of cancer inclusion, the authors feel intuitively fragmented, noncompact, and unsystematic in their introduction to the relationship of NUMB to various carcinomas. With the amount of literature allowed, the scope of cancer can be further narrowed at the time of writing.

In reference to your comment, the authors would like to justify our decision not to focus on a few representative tumors to show the functions of NUMB. The idea of this review is to expose the multifunctionality of NUMB in a general overview, without focusing on any single tumor type. This allow us to demonstrate both the wide variety of tumors on which NUMB could have an effect and that this effect could be tissue-specific.

  1. Almost all titles are centered on NUMB, but the content under the title also contains a large number of descriptions of the relationship between NUMB and diseases. Therefore, some titles may be further modified.

According to your suggestion to link disease descriptions with NUMB in titles, we have changed some of the titles.

  1. Line 13 According to the previous literature, I think the author cannot support this statement.

In response to your comment, we simplified the sentence on line 13 by removing a statement not related to the previous sentence to better align with the knowledge presented in the review.

  1. Line 158 The word "Thus" is inappropriate.

We replaced the inappropriately used word "Thus" with "In more detail" (line 178).

  1. The meanings of arrows in Figure 3 shall be marked, and the key steps shall be indicated with symbols.

As recommended, we added a legend to the arrows in Figure 3. To avoid making the figure difficult to visualize, we indicated in the legend the different signaling pathways affected by NUMB, including the following text: "NUMB inhibits NOTCH signaling by interacting with NICD. Additionally, the interaction between NUMB and MDM2 increases p53 stability, allowing transcription of genes related to DNA-damage repair. Furthermore, the PRRL NUMB isoform promotes ERK/MAPK signaling and the formation of a SMAD complex, leading to transcription of genes related to EMT. In addition, NUMB promotes the ubiquitination and subsequent degradation of both NUMB and ITCH through GLI1."

  1. Lines327-329 It was written in the original that NUMB promoted NOTCH ubiquitination and NICD degradation by recruiting ITCH, but this relationship was not directly reflected in Figure 3.

Based on your indication, we also modified Figure 3 to add the step where NUMB promotes NOTCH ubiquitination and NICD degradation through ITCH recruitment..

  1. Lines 336-337 The role is not specific.

To clarify the role of NUMB in cardiogenesis, we replaced the previous sentence with the following:  ”NUMB also plays a role in the proliferation of cardiomyocytes and trabecular morphogenesis through its interaction with NOTCH1’. This clarifies the specific function of NUMB in this process (lines 340-341).

  1. Line 337 The word "Otherwise" is inappropriate.

We replaced the inappropriately used word "Otherwhise" with "However" (line 342).

Sincerely,

José Manuel García-Heredia ([email protected])

Sara M. Ortega-Campos        ([email protected])

Reviewer 2 Report

The review article proposed by Ortega-Campos and García-Heredia summarizes the current knowledges about the multiple roles of NUMB protein. The manuscript lists the biological processes and the interaction partners of NUMB, mentioning almost all the physiological regulatory mechanisms (and their deregulations in cancer) that modulate NUMB function.

Overall, this review is clear. Few corrections to the manuscript are necessary (some are reported in the comments below) before re-submission.

Some sentences are not very concise. I suggest to the authors, where possible, to avoid breaking up a concept with many indents to make reading more fluid.

My specific comments follow:

·         Just some checklist for style: 

o   Some commas are missing throughout the paper (especially when listing more than three items in the same sentence);

o   Please, when possible, use a wider vocabulary to avoid repetitions (ex. Lines 111-112: modified… modifying);

o   Please, uniform “Drosophila” and “Drosophila”;

o   Line 26: to make these advances;

o   Lines 43/44: CSC (Cancer Stem Cell) please, insert the abbreviation after the full name;

o   Line 75: SNP please, specify the full name of this abbreviation;

o   Line 104: Exons and domain sizes are not to scale; they are simplified for facilitating visualization.

Author Response

Dear Reviewer,

We would like to thank you for your time and effort in reviewing our manuscript. We appreciate the positive comments you have made about the clarity and the comprehensiveness of the information summarized in our review. We have worked on your comments and have attempted to bring you a version with all the modifications you have suggested to improve our paper. We would be delighted to receive further comments that will allow us to improve this work.

Below, we provide responses to each of your comments.

Some sentences are not very concise. I suggest to the authors, where possible, to avoid breaking up a concept with many indents to make reading more fluid.

We have carried out a deep revision of the text, to improve its understanding.

Regarding your specific comments:

  1. Some commas are missing throughout the paper (especially when listing more than three items in the same sentence);

As can be observed, missing or extra commas in the paper have been revised.

  1. Please, when possible, use a wider vocabulary to avoid repetitions (ex. Lines 111-112: modified… modifying);

Our manuscript has been checked for possible repetitions as in the example you indicated.

  1. Please, uniform “Drosophila” and “Drosophila”;

Based on your advice, the term Drosophila has been italicized.

  1. Line 26: to make these advances;

We have modified the entire sentence to correct it, although the use of “making” is maintained according to English revisions: “Investigating proteins that are involved in human diseases and play a role in development and cell fate could be key to making connections and advances by understanding how their presence and function modify cell behavior”.

  1. Lines 43/44: CSC (Cancer Stem Cell) → please, insert the abbreviation after the full name;

The abbreviation CSC has been inserted after Cancer Stem Cell as you suggested (lines 46-47).

  1. Line 75: SNP → please, specify the full name of this abbreviation;

The full name SNP: Single Nucleotide Polymorphism has been specified, as you pointed out (line 80).

  1. Line 104: Exons and domain sizes are not to scale; they are simplified for facilitating visualization.

The grammatical error in the figure 1 caption has been corrected: "Please note that the sizes of exons and domains in the diagram are not to scale and are simplified for clarity"

Sincerely,

José Manuel García-Heredia ([email protected])

Sara M. Ortega-Campos        ([email protected])

Reviewer 3 Report

As a multi-task protein involved in determining cell function and cell fate from early growth and development, NUMB has significant implications for studying the mechanisms of human diseases such as cancer and Alzheimer's disease.The topic of this review is meaningful.This study introduces NUMB and its related functions, and expounds the related clinical significance, which shows the important role of NUMB in maintaining homeostasis from many aspects.However, the content of this article is too simple and the logic is poor. The author only briefly summarizes the existing research and lacks in-depth thinking and exploration.There are many areas in need of improvement, which are reflected in the following aspects.

1. In the introduction to the article, the author mentioned how the NUMB function would be involved in the development of pathology such as Alzheimer's disease and cancer, but in a later article the author referred to the relationship between NUMB and Alzheimer's disease and cancer in a smaller space, deviating from the subject. Moreover, when introducing the mechanism of NUMB and disease, the literature reference is less, the lack of a certain amount of evidence, can not effectively convince readers.

2. About the function of NUMB in cells, the author spent a lot of space on the fifth part 'NUMB as an endocytic protein'. In contrast, other functions are a little sketchy in the article. The author should make a more detailed classification of other functions. At the same time, the fifth part also lacks the extraction of the literature.

3. The logic of the article is poor, most of which are the author's simple list of research results, lacking his own summary and thinking. For example, the 'clinical implications of NUMB' part lacks a summary of the author treatment methods. There is no obvious logical relationship between NUMB and disease. The discussion is too simple and not deep enough.Such without clear and convincing proof's articles are of little value to readers.

4. In this paper, there are few relevant research literatures in recent years, which can not well reflect the latest achievements of NUMB, and the timeliness and innovation are insufficient.

5. There are a large number of long sentences in the fourth, fifth and sixth parts, which are difficult to understand. The lack of concise extraction of the literature will increase readers' reading difficulties.

6. Some pictures lack illustrations, which adds obstacles to readers' reading.

    The overall content and framework of this paper still exist some problems, and lack of logic and innovation. Therefore, the reviewer recommended rejection.

Author Response

Dear Reviewer 3,

First of all, the authors would like to thank you for taking the time and dedication necessary to carry out such a careful review. We are deeply sorry that it did not meet your expectations and that you proposed it for rejection. Your comments and needs for improvement of our manuscript have been highly valued. After a major revision and evaluation of our manuscript following your advice, we are sending you this new version, which we hope will finally meet your standards and convince you to approve the publication of our manuscript. As we know that this is an arduous task, we will be happy to receive any further suggestions for improvement that you may think necessary.

Below, we provide responses to each of your comments:

  1. In the introduction to the article, the author mentioned how the NUMB function would be involved in the development of pathology such as Alzheimer's disease and cancer, but in a later article the author referred to the relationship between NUMB and Alzheimer's disease and cancer in a smaller space, deviating from the subject. Moreover, when introducing the mechanism of NUMB and disease, the literature reference is less, the lack of a certain amount of evidence, can not effectively convince readers.

Regarding your first comment, the bibliographical support has been strengthened to introduce the mechanism between NUMB and the disease. In this way, the authors hope to provide sufficient evidence for readers to be effectively convinced of the quality of our manuscript. In addition, the introduction was expanded by writing more about the relationship between NUMB and disease as well as its therapeutic potential.

  1. About the function of NUMB in cells, the author spent a lot of space on the fifth part 'NUMB as an endocytic protein'. In contrast, other functions are a little sketchy in the article. The author should make a more detailed classification of other functions. At the same time, the fifth part also lacks the extraction of the literature.

The authors would like to clarify that the part describing the endocytic function of NUMB is the longest section because it is the best known and the one for which the most information is available. However, some parts have been expanded, such as the part on NUMB as a therapeutic tool.

  1. The logic of the article is poor, most of which are the author's simple list of research results, lacking his own summary and thinking. For example, the 'clinical implications of NUMB' part lacks a summary of the author treatment methods. There is no obvious logical relationship between NUMB and disease. The discussion is too simple and not deep enough. Such without clear and convincing proof's articles are of little value to readers.

In order to carry out this review, the authors first set out to produce a manuscript containing the most relevant information known about NUMB. To do this, we searched for articles using the term NUMB in a bibliographic search engine. From there, we thoroughly analyzed the extracted information and tried to summarize the information on NUMB from its characteristics as a gene/protein to all the functions it plays within the cell in the maintenance of homeostasis. We pointed out the implications of these functions in the development of various pathologies, thus justifying its potential as a therapeutic tool. In order to address the problem, we have revised the entire text to make it more logical in structure. We have also revised the clinical part of our manuscript, adding new literature to reinforce the justification for the therapeutic potential of NUMB.

  1. In this paper, there are few relevant research literatures in recent years, which can not well reflect the latest achievements of NUMB, and the timeliness and innovation are insufficient.

Based on your suggestion for improvement, we have carried out a new bibliographic search to provide more references from recent years to demonstrate the innovation and relevance of our proposal.

  1. There are a large number of long sentences in the fourth, fifth and sixth parts, which are difficult to understand. The lack of concise extraction of the literature will increase readers' reading difficulties.

We have thoroughly checked the text for sentences that are too long and could make the text difficult to read.

  1. The authors are not sure what the reviewer means by “Some pictures lack illustrations, which adds obstacles to readers' reading”. However, we have modified Figure 3 to include a legend explaining the arrows that appear. Additionally, we indicated in the legend the different signaling pathways affected by NUMB, including the following text: "NUMB inhibits NOTCH signaling by interacting with NICD. Additionally, the interaction between NUMB and MDM2 increases p53 stability, allowing transcription of genes related to DNA-damage repair. Furthermore, the PRRL NUMB isoform promotes ERK/MAPK signaling and the formation of a SMAD complex, leading to transcription of genes related to EMT. In addition, NUMB promotes the ubiquitination and subsequent degradation of both NUMB and ITCH through GLI1."
  2. The overall content and framework of this paper still exist some problems, and lack of logic and innovation.

Thank you for your feedback. We apologize for any problems or lack of logic and innovation in the content and framework of the previous version of the paper. We have worked to address these issues in order to improve the paper. We hope to have satisfied, with this reviewed version, at least some of your comments and suggestions.

Finally, the authors would like to add that the use of the English language has been carefully revised and appropriate changes have been made.

Sincerely,

José Manuel García-Heredia ([email protected])

Sara M. Ortega-Campos        ([email protected])

Round 2

Reviewer 1 Report

Comments to the Author(s)

Title:The multitasker protein: A look at the multiple capabilities of NUMB.

In the latest manuscript, you revised the article substantially in response to the questions that I asked for the first time. I think the revised manuscript is of a much improved quality than before and has met the publication criteria of this journal. Therefore I recommend agreeing to accept the manuscript.

Reviewer 3 Report

Accept.